# Testing the Effect of Cue Consistency on the Past Behavior–Habit–Physical Activity Relationship

**DOI:** 10.3390/bs14060445

**Published:** 2024-05-24

**Authors:** Daniel J. Phipps, Martin S. Hagger, David Mejia, Kyra Hamilton

**Affiliations:** 1School of Applied Psychology, Griffith University, Mount Gravatt, QLD 4122, Australia; mhagger@ucmerced.edu (M.S.H.); kyra.hamilton@griffith.edu.au (K.H.); 2Faculty of Sport and Health Sciences, University of Jyväskylä, 40014 Jyväskylä, Finland; 3Department of Psychological Sciences, University of California, Merced, CA 95343, USA; 4Health Sciences Research Institute, University of California, Merced, CA 95343, USA; 5College of Health and Human Performance, University of Florida, Gainesville, FL 32611, USA; david.mejia@ufl.edu

**Keywords:** habit, physical activity, cues, dual-process model

## Abstract

Behavior performed in the presence of consistent cues is a core element for successful habit development, with the repeated presence of consistent cues facilitating the activation of automatic responses in future. Yet, little is known about the effects of different cue types on habit. Using a two-wave prospective PLS-SEM model with a sample of 68 undergraduate students, we assessed the mediating effects of habit on the past-behavior-to-physical-activity relationship, and how the mediating effects of habit were moderated by the consistent presence of different forms of cues. Habit mediated the effects of past behavior on physical activity, with a significantly stronger mediating effect of habit in those reporting undertaking physical activity at the same time of day, doing the same activity, and in the same mood. Consistent place, people, and part of routine did not moderate the effects of habit. The results provide formative evidence for a key assertion of the habit theory that consistent contextual and internal cues are a cornerstone of habitual development and action, but they also indicate the importance of examining different forms of cues and their impact on the formation and enaction of habits as some cues may be more relevant than others.

## 1. Introduction

Research identifying the determinants of health behavior has largely been grounded in social cognition theories. These theories assume that behavior is the consequence of a deliberative, reasoned process whereby an individual considers beliefs relevant to their current environment and makes a decision as to whether to act [1]. Evidence using such theories, including prototypical models like the theory of planned behavior [2] and the health action process approach [3], has generally demonstrated expected model effects in predicting behavior, as shown through meta-analysis [4,5,6]. Yet, despite the non-zero effects, such theories have come under increasing criticism as reports consistently show that only a modest portion of variance in behavior is accounted for by social cognition constructs, and changes in intention are not always matched with equivalent changes in actual behavior [7]. One prominent explanation for this shortfall of social cognition models in explaining behavior is that people’s actions are unlikely to be determined solely by reasoned, deliberative processes. Instead, it is likely many day-to-day, frequently performed behaviors fall under the control of highly efficient, automatic processes [1,8,9].

Acknowledging the likelihood of automatic constructs as potential determinants of behavior, a dominant line of research in theory development has focused on testing more integrated models of behavior that include constructs purported to assess automatic behavior, alongside beliefs based on intentional variables that represent social cognition [10,11,12,13]. One common construct used to represent automatic behavior in studies testing dual process integrated models is habit [14,15,16,17]. Traditional definitions of habit are generally based on the relative frequency of a behavior [18]. However, more contemporary theorical perspectives view habit as the degree to which a behavior is enacted automatically [14], that is, without the need for significant conscious input, such as the consideration of potential benefits or outcomes of a behavior before making a decision to act.

The basis of such automatic actions is drawn from the overall conceptualization of the habit construct as a stimulus–response effect based upon the connection between cues, contexts, and actions in associative memory [14,19,20]. Thus, a key thread in contemporary habit theory is that the development and enactment of more automatically performed actions should be contingent on the presence of consistent environmental or internal contexts in which a behavior is repeatedly performed [21,22,23], that is, a habit is formed when a behavior is repeated in the presence of a stable internal (e.g., mood) or external cue (e.g., place, time, and people), and the behavioral response becomes linked to the cue in associative memory. Once the cue and behavioral response become associated in memory, encountering that cue or context again should activate the learned stimulus–response effect and, thus, be sufficient to trigger the action chains encouraging behavioral enactment with little or no conscious deliberation [19]. Meta-analytic evidence supports the effect of habit on behavior alongside social cognition constructs [13,24]. However, theoretical debate continues around the contextual or cue-based factors that influence the development and enactment of habitual behaviors.

Despite theoretical assertions of the potential role of cue-consistent contexts in the development of habits, empirical evidence supporting such effects is limited. For example, a seminal study in habit development found that the self-selection of cues for a health behavior was associated with the development of stronger habits for that behavior over time [21]. Yet since this study, there has been a relative dearth of research investigating the effects of different forms of cues on habit development and enactment. Such a contention may be particularly important, as some habit theories have speculated that different forms of cues may have differential impacts on how likely they are to result in the development of habits [25,26]. Such research suggests that cues that are more likely to be salient and easily noticed (e.g., going for a walk after dinner) should be more likely to result in the development of habits than cues that are less salient or require conscious monitoring or input (e.g., going for a walk each day at 6 p.m. requires a person to be consciously aware of the time) [27].

While limited, one example of research that empirically differentiates the effect of cue types on habits is presented in the predominant measure of cue consistency, which breaks down potential consistent factors associated with the development of behavior into habit as enacting the behavior at a consistent time, place or mood, at the same part of one’s routine, as the same activity, or with the same people [28]. Yet, other than the original test of this measure in a cross-sectional study associating different forms of cue consistency and physical activity, little research discerning the differential effects of cue types has been undertaken. This study by Pimm et al. showed a significant zero-order correlation between habit strength and the enactment of a behavior at a consistent time of day, with consistent people, as a consistent activity, at the same part of one’s routine, in the same location, and in a consistent mood; while consistent time, people, and part of routine were associated with self-reported behavior, but consistent activities, place, and mood were not [28]. Such findings provide preliminary evidence in support of the role of consistent cues in habit development, but they also indicate potential differential effects in how each form of consistent cue may translate into actual behavior.

Further, the interpretation of effects in the research by Pimm et al. is notably limited by the study’s cross-sectional design. Specifically, a fundamental element of habit theory is that habits should mediate the relationship between past and future behaviors, that is, habits are theorized to form on the basis of past behavioral experiences occurring in the presence of stable cues. Then, once formed, the encountering of these cues should activate the associated habitual response, thus increasing the frequency of a behavior. A fundamental issue with testing the relationship between cues, habits, and behaviors using cross-sectional data is that it is difficult to distinguish between different effects, as a cross-sectional relationship between habit and behavior could be indicative of both the effect of past behavior on habit development or the effects of habit in encouraging the behavior. Thus, while previous research and theory indicates a link between habits, cues, and behaviors, the use of longitudinal or prospective designs is required to provide more in-depth tests of habit theory, such as the extent to which habit mediates the past behavior–future behavior relationship and how different forms of consistent cues may influence both the development and enaction of habits.

## 2. The Current Study

In the current study, we sought to investigate whether the effects of past behavior on prospectively measured behavior were mediated by habit, and the extent to which the mediating effect of habit is dependent on engaging in the behavior in the presence of consistent cues or contexts. In this study, we tested a key health behavior, engaging in physical activity according to current guidelines. Specifically, we hypothesized that habit would mediate the effects of past behavior on future behavior, as past behavior would predict habit, while habit would, in turn, predict prospectively measured behavior. Further, we hypothesized that, in the presence of consistent cues (i.e., time of day, activity, people, part of routine, place, and mood), the effects of past behavior on habit and of habit on prospectively measured behavior would be stronger, compared to when physical activity behavior was enacted in the presence of inconsistent cues. In contrast, we expected that when behavior was reported as occurring in the presence of inconsistent cues, the effect of past behavior on future behavior not accounted for by habit would be stronger, as a continuing behavior would likely be accounted for by non-automatic constructs.

## 3. Methods

### 3.1. Study Setting

Data were collected in a prospective study with a two-week time lag. Participants were recruited for course credit from a U.S. university between March and June 2023, and were eligible to participate if they did not have any conditions or injuries which prohibited them from being physically active.

### 3.2. Participants and Study Size

At baseline, 138 participants completed measures of past behavior in an online survey hosted on the Qualtrics platform. Participants were then emailed two weeks later to complete measures of habit, cue consistency, and prospectively measured behavior. However, 70 participants did not respond to requests for follow-up data. Thus, the final sample consisted of 68 participants (*M* age = 20.82, *SD* age = 3.10, 58 female, 24 male, 2 non-binary). The attrition analyses indicated that the included sample did not significantly differ from those who did not respond to requests for follow-up data in terms of age, *t*(136) = 2.60, *p* = 0.338, *d* = 0.07, or gender, χ^2^(2) = 4.78, *p* = 0.092. All procedures were approved by the University of California Merced Institutional Review Board, reference no. UCM2022-77.

### 3.3. Variables and Measurements

All survey items are presented in Table 1.

**Past Behavior**. Past behavior was assessed using three items (e.g., In the past year, how often did you meet the physical activity guidelines each week?), with each item scored on a 7-point Likert-type scale (e.g., (1) Never to (7) Always) [29].

**Habit**. Habit was assessed using the behavioral automaticity subscale of the self-reported habit index [30,31]. The scale consisted of four items (e.g., Meeting the physical activity guidelines each week is something I do without having to consciously remember), with each item scored on a 7-point Likert scale anchored from (1) Strongly Disagree to (7) Strongly Agree.

**Cue Consistency**. Cue consistency for physical activity was assessed using a modified version of the Pimm cue consistency scale [28], asking participants to record whether they engaged in physical activity at the same time of day, in the same mood, with the same people, at the same place, as the same part of their routine, or doing the same activity. Responses were scored on a 7-point Likert scale anchored from (1) Strongly Disagree to (7) Strongly Agree.

**Behavior**. Physical activity behavior was assessed using three items (e.g., In the two weeks, how often did you meet the physical activity guidelines each week?), with each item scored on a 7-point Likert-type scale (e.g., (1) Never to (7) Always) [29].

### 3.4. Statistical Methods

Data were fitted to a bootstrapped PLS-SEM model with 10,000 iterations using the SmartPLS 4 software [32], with the survey items for past behavior, habit, and physical activity used as indicators for latent constructs. In the multi-group moderation analysis, for each cue in the cue consistency scale, responses above the scale’s midpoint (i.e., a rating of agree or strongly agree that a cue was consistent) were coded as enacting the behavior with a consistent cue, while responses at or below the scale’s midpoint (i.e., a rating of strongly disagree, disagree, or neither agree nor disagree that a cue was consistent) were categorized as indicative of the behavior being inconsistently enacted with each cue. The estimates between high and low cue-consistency groups were compared using a bootstrapped multi-group analysis with 10,000 iterations. All survey items required a response; thus, no missing data analysis was conducted.

## 4. Results

The zero-order correlations, reliability statistics, and descriptive statistics for each variable are presented in Table 2. All scales showed good reliability, with all items loading significantly upon their expected factors (all *p*-values < 0.001). The full item loadings for each model are also available in Appendix A.

In the total sample, the model predicted 47.3% of the variance in behavior, with past behavior predicting future behavior both directly and indirectly through the mediating effect of habit. The model for the total sample is presented in Figure 1.

Despite the expected stronger effects of past behavior on habit in those reporting consistent cues, the moderating effect of cues did not reach the statistical significance threshold for any given cue. However, the effect of habit on behavior was stronger in those who reported undertaking physical activity at a consistent time of day, in a consistent mood, and as a consistent activity. Consequently, the mediating effect of habit between past and future behaviors was significantly stronger in those undertaking physical activity at a consistent time of day, in a consistent mood, and as a consistent activity. In contrast, the effect of past behavior on prospectively measured behavior not accounted for by habit was stronger in those reporting undertaking physical activity at an inconsistent time, as an inconsistent activity, and in an inconsistent mood. All parameter estimates are presented in Table 3.

## 5. Discussion

The current study sought to test whether the mediating effect of habit on the past-behavior-to-future-behavior relationship was moderated by the presence of consistent cues or contexts for the behavioral enactment of a key health behavior—engaging in physical activity. In line with our hypotheses, habit mediated the effects of past behavior on future behavior in the whole-sample model. Such a finding is consistent with previous research [10,13,33], as well as theories postulating that habit formation is likely dependent on past behavioral experiences, and once formed, habits promote the continuation of a behavior through the activation of stimulus–response-triggered actions [14]. Further in line with the hypotheses, the effect of past behavior on prospectively measured physical activity was not entirely accounted for by habit, as past behavior remained a significant predictor of behavior. These results are also in line with integrated models, as the influence of past behavior on future behavior is likely to also be modeled by other, non-habitual factors, such as intention or affect, which have also demonstrated significant effects on physical activity in prior research [34,35]. 

We also tested the hypothesis that the mediating effect of habit on the past-behavior-to-future-behavior relationship should be stronger in the presence of consistent cues, that is, the development of habit should be contingent on the frequent co-occurrence of a behavior and a cue, allowing for the cue and the action to become linked in associative memory. Once this link is formed, the sustained presence of the cue, either externally or internally, should increase the likelihood of the behavior occurring habitually due to the stimulus–response effect resulting from the perception of the cue. The results in this regard were partially in line with expectations, as the indirect effect of past behavior on prospectively measured behavior via habit was significantly stronger in participants reporting engaging in physical activity at the same time, as the same activity, and in the same mood. However, in contrast to our hypotheses, engaging in physical activity in the same place, with the same people, and as the same part of one’s routine did not significantly moderate the mediating effect of habit, despite moderation effects in the expected direction. These results present a unique contribution of this research, indicating that while cues are likely an important element in promoting habit development and enaction, not all forms of cues appear to have an equal impact in this regard. While some differences in the role of different cues are expected, the current findings may be of particular interest as, based on habit theory, one may expect more salient cues (e.g., routine) to be stronger mediators of this effect than cues such as time or mood [25,27]. Yet, the current results do not indicate this to be the case, as time but not routine moderated the effects of past behavior via habit on physical activity. While future research is needed to confirm these findings, they may have notable implications for intervention, particularly since many habit formation and disruption strategies to date are based upon the formation or removal of cues to alter habitual behaviors [21,25,36,37,38,39]. 

It is also important to consider that, upon a closer examination of these mediating effects, even where a significant mediating effect was observed, the results are not totally in line with habit theory. Specifically, based on theory, one would expect past behavior to be a stronger predictor of habit in those who reported undertaking the behavior in the presence of consistent cues, given that the stability of these cues should be considered a theoretical prerequisite for habit development [19,40]. Yet, in the current data, while the past-behavior-to-habit pathway was stronger when cues were consistent in each case, this moderating effect was small and fell short of the criteria for statistical significance for each cue. Thus, in each case, while the mediating effect of habit was stronger in the presence of consistent time, mood, and activity cues, this significant effect could mostly be attributed to these cues showing a strong moderating effect on the habit-to-behavior pathway. 

On the surface, such a finding may seem to contest the theoretical assumption that consistent cues are necessary for habit development. However, there are also several other plausible explanations for the observed pattern of findings. For example, in light of the trend of the stronger effects of past behavior on habit in the presence of all cues, it is also plausible that the current effects may be explained as consistency, although not in any one cue, is required to facilitate habit development with strong effects [27]. For example, it may be that simply being active at the same time is insufficient for a habit to form [27], and one may also need to be active at the same time, doing the same activity, and in the same place for that activity to become habitual. However, due to the modest sample in the current study, we were unable to test the numerous potential moderating effects of different cue types, and future research with larger samples is required. Alternatively, given that habits are highly stable over time [41,42], and the design of the current study did not require any change or alternation of behavior, it may be that consistent cues had only small effects on exercise habits as, for many participants, such habits were likely to have already been firmly established. While such explanations are plausible, future research is needed to investigate the effect of cue consistency in contexts where habit change or the formation of new habits is likely to occur (e.g., moving to college and legislative changes) [42,43,44].

We also observed a stronger effect of past behavior on prospectively measured behavior when behavior was not enacted in the presence of consistent time, mood, and activity cues. While this effect is somewhat inconsistent with data on past behavior by frequency measures of the habit construct [45], it is likely that in the current study, the residual effect of past behavior on future behavior beyond habit reflects non-habitual constructs such as intention. Thus, extending on the findings regarding habit, it is likely that while consistent cues encourage habitual action, acting in inconsistent settings and contexts requires more intensive cognition, thus increasing the likelihood of making a considered decision [43,44,46,47]. However, as intentional and belief-based constructs were not included in the tested model, such an explanation remains speculative. Future research may also seek to directly assess the effect of inconsistent cues on the role of intentions on behavior alongside habits. 

## 6. Strengths, Limitations, and Future Directions

While the current study presents a valuable contribution to research on integrated behavior change models seeking to understand when habitual processes likely occur, it is inherently not without limitations. Firstly, the current study made use of brief self-reported scales for both past and future behaviors. While similar scales have shown acceptable evidence for their validity in terms of correlations with observational measures or more intensive survey designs [48], self-reported measures of behavior, by their nature, are at risk of biased or inaccurate responding. Future research may seek to replicate these findings using more objective, observational measures of behavior, such as heart rate monitors, pedometers, or accelerometers. Further, the current study made use of a prospective, two-wave design. While the prospective design allows for a better test of mediating effects, it does not allow for assertions of causality. Future research may seek to test similar models using longitudinal designs, for example, by assessing habit development alongside cue consistency in the wake of major context changes where new habits are likely to be developing.

## 7. Conclusions

The current study sought to test the extent to which the presence of consistent cues increased the strength of the mediating effect of habit on the relationship between past behavior and future behavior with regard to physical activity. As expected, the indirect effect of past behavior on future behavior via habit was stronger when behavior was enacted at the same time of day, in the same mood, and as the same activity, although no moderating effect was observed for consistent place, people, or part of routine. The current findings support habit theory, while also providing additional evidence highlighting the potential efficacy of cue-based habit change strategies, particularly those focusing on time, activity, or mood. However, future longitudinal research is required to investigate how cues may affect habit development, particularly how cues may co-occur and in contexts where significant habit change or development may be expected.

## Figures and Tables

**Figure 1 behavsci-14-00445-f001:**
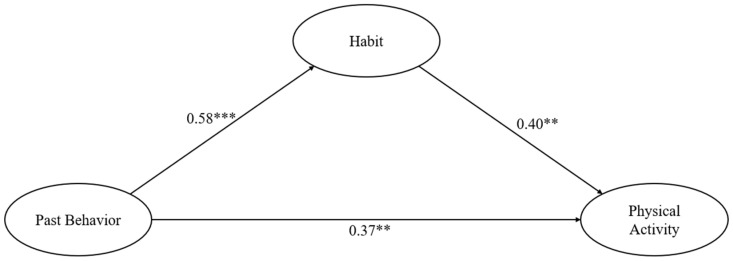
The model predicting physical activity through past behavior and habit. Note. ** indicates *p* < 0.010, and *** indicates *p* < 0.001.

**Table 1 behavsci-14-00445-t001:** Self-reported scales for each construct.

Items	Response Options
In the past year, to what extent did you meet the physical activity guidelines each week?	(1) A small extent to (7) A large extent
In the past year, how often did you meet the physical activity guidelines each week?	(1) Never to (7) Always
In the past year, I met the physical activity guidelines each week?	(1) False to (7) True
Meeting the physical activity guidelines each week is something…	
… I do automatically.	(1) Strongly Disagree to (7) Strongly Agree
… I do without having to consciously remember.	(1) Strongly Disagree to (7) Strongly Agree
… I do without thinking.	(1) Strongly Disagree to (7) Strongly Agree
… I start to do before I realize I’m doing it.	(1) Strongly Disagree to (7) Strongly Agree
Each time I start to engage in moderate or vigorous physical activity ….	
… it is the same time of the day.	(1) Strongly Disagree to (7) Strongly Agree
… I am around the same people.	(1) Strongly Disagree to (7) Strongly Agree
… I do the same type of activity.	(1) Strongly Disagree to (7) Strongly Agree
… I am in the same part of my routine.	(1) Strongly Disagree to (7) Strongly Agree
… I am in the same place.	(1) Strongly Disagree to (7) Strongly Agree
… I am in the same mood.	(1) Strongly Disagree to (7) Strongly Agree
In the two weeks, to what extent did you meet the physical activity guidelines each week?	(1) A small extent to (7) A large extent
In the two weeks, how often did you meet the physical activity guidelines each week?	(1) Never to (7) Always
In the two weeks, I met the physical activity guidelines each week?	(1) False to (7) True

**Table 2 behavsci-14-00445-t002:** Descriptive statistics and zero-order correlations between all study variables.

	1	2	3	4	5	6	7	8	9
1. Past Behavior	-								
2. Activity Consistency	0.21	-							
3. Mood Consistency	0.03	0.34 **	-						
4. People Consistency	0.29 *	0.53 ***	0.45 ***	-					
5. Place Consistency	0.23	0.72 ***	0.46 ***	0.52 ***	-				
6. Time Consistency	0.36 **	0.57 ***	0.36 **	0.56 ***	0.59 ***	-			
7. Routine Consistency	0.26 *	0.86 ***	0.37 **	0.65 ***	0.75 ***	0.65 ***	-		
8. Habit	0.58 ***	0.25 *	0.18	0.41 ***	0.24 *	0.35 **	0.30 *	-	
9. Behavior	0.60 ***	0.11	0.10	0.27 *	0.14	0.44 ***	0.19	0.62 ***	-
Mean	3.51	5.03	3.79	4.15	4.99	4.15	4.69	3.19	3.23
*SD*	1.60	1.56	1.60	1.73	1.58	1.72	1.66	1.57	1.77
Cronbach’s α	0.940	-	-	-	-	-	-	0.967	0.977
AVE	0.962	-	-	-	-	-	-	0.977	0.985
Square Root of AVE	0.980	-	-	-	-	-	-	0.988	0.992

Note. * indicates *p* < 0.05, ** indicates *p* < 0.010, and *** indicates *p* < 0.001. AVE refers to average variance extracted. Numbers in the column header row refer to each of the numbered constructs within row names.

**Table 3 behavsci-14-00445-t003:** Parameter estimates for the models predicting physical activity.

	Model SRMR	Past Behavior to Behavior		Past Behavior to Habit		Habit to Behavior		Habit Mediation	
		β	*p*	β	*p*	β	*p*	β	*p*
Whole Sample	0.029	0.37	0.004	0.58	<0.001	0.40	0.003	0.23	0.008
Activity									
Consistent	0.033	0.26	0.069	0.59	<0.001	0.50	0.001	0.30	0.006
Inconsistent	0.056	0.82	<0.001	0.54	0.074	0.02	0.917	0.01	0.939
*p*-value of difference		0.025		0.494		0.035		0.046	
Mood									
Consistent	0.037	0.14	0.420	0.61	<0.001	0.73	<0.001	0.44	<0.001
Inconsistent	0.037	0.57	<0.001	0.58	<0.001	0.13	0.483	0.08	0.511
*p*-value of difference		0.045		0.437		0.009		0.018	
Routine									
Consistent	0.036	0.42	0.018	0.65	<0.001	0.33	0.083	0.22	0.111
Inconsistent	0.057	0.17	0.458	0.35	0.420	0.52	0.010	0.18	0.432
*p*-value of difference		0.817		0.273		0.754		0.476	
People									
Consistent	0.039	0.42	0.035	0.64	<0.001	0.33	0.131	0.21	0.158
Inconsistent	0.042	0.34	0.064	0.47	0.006	0.47	0.004	0.14	0.150
*p*-value of difference		0.616		0.197		0.691		0.484	
Time									
Consistent	0.035	0.08	0.624	0.64	<0.001	0.70	<0.001	0.44	<0.001
Inconsistent	0.051	0.60	<0.001	0.50	0.008	0.09	0.649	0.04	0.687
*p*-value of difference		0.013		0.243		0.007		0.008	
Place									
Consistent	0.032	0.31	0.042	0.58	<0.001	0.45	0.005	0.26	0.015
Inconsistent	0.065	0.63	0.001	0.52	0.015	0.16	0.396	0.08	0.521
*p*-value of difference		0.079		0.434		0.114		0.130	

Note: *p*-value of difference refers to the *p*-value for the bootstrapped difference test for the strength of the parameter estimates between those with a consistent or inconsistent cue. SRMR refers to the standardized root mean square residual, where scores below 0.08 indicate a good model fit.

## Data Availability

Data presented in this study is available at https://doi.org/10.17605/OSF.IO/THS27.

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
