# Peer review of "Testing the Effect of Cue Consistency on the Past Behavior–Habit–Physical Activity Relationship"

_behavsci, 2024, doi:10.3390/bs14060445_

Round 1

Reviewer 1 Report

Comments and Suggestions for Authors

The manuscript explores the role of consistent cues in the development of habits and their impact on physical activity behavior. The study tested the hypothesis that habit mediates the effect of past behavior on future behavior and that the mediating effect of habit is dependent on engaging in behavior in the presence of consistent cues or contexts. The study’s findings suggest that the presence of consistent cues enhances the mediating effect of habit on the relationship between past behavior and future behavior for physical activity, emphasizing the potential efficacy of cue-based habit change strategies. In summary, the manuscript has significant potential, but I recommend addressing several key issues before considering its publication:

1) The introduction presents a coherent and relevant review of previous studies, effectively identifying a gap in the existing literature. The focus on the need for longitudinal or prospective designs for a deeper evaluation of habit theory is crucial. However, although the influence of different forms of consistent cues on the development and enactment of habits is mentioned, it would be useful to specify which forms these cues will take and why they are expected to have a differential impact.

2) I appreciate the use of PLS-SEM to analyze the structural relationships in the study, which is suitable given the nature of the hypotheses posed. However, it is crucial to include details about the reliability of the scales used to measure each construct. This would not only increase the transparency of the study but also allow readers to evaluate the robustness of the results. I recommend providing reliability coefficients such as Cronbach's alpha, Jöreskog's rho, or McDonald's omega for each scale used. Additionally, it would be beneficial to discuss construct validity through convergent and discriminant validation, providing values for factor loadings, average variance extracted (AVE), and the maximum shared variance (MSV) or the square root of the AVE compared with inter-construct correlations. Including these methodological details would not only enrich the methodology section but also strengthen the credibility of the study's findings.

3) Moreover, I note a significant omission in the lack of reporting of model fit measures. It is essential to include fit indices such as the Normative Fit Index (NFI), Chi-square (Chi²), and the Standardized Root Mean Square Residual (SRMR), among others. These indices are fundamental to evaluating the adequacy of the proposed measurement and structural models, thus allowing readers and reviewers to judge the quality and reliability of the obtained results.

4) In the methodology section, the use of PLS-SEM to examine the structural relationships between latent variables is detailed. However, to improve clarity and comprehensibility of the study for readers, I strongly recommend that the authors include a visual graph of the proposed structural relationships. A diagram showing the relationships, including path coefficients and significances, would facilitate the interpretation of the models and allow readers to more effectively visualize the dynamics between the variables. Furthermore, this type of graphical representation is a standard practice in the presentation of results in studies applying SEM and would be a valuable addition to complement the textual description of the analysis.

Author Response

Reviewer 1

Reviewer’s Comment: The manuscript explores the role of consistent cues in the development of habits and their impact on physical activity behavior. The study tested the hypothesis that habit mediates the effect of past behavior on future behavior and that the mediating effect of habit is dependent on engaging in behavior in the presence of consistent cues or contexts. The study’s findings suggest that the presence of consistent cues enhances the mediating effect of habit on the relationship between past behavior and future behavior for physical activity, emphasizing the potential efficacy of cue-based habit change strategies. In summary, the manuscript has significant potential, but I recommend addressing several key issues before considering its publication:

Author’s Reply: We thank the reviewer for their positive feedback and thoughtful consideration of our manuscript. We have endeavoured to make the changes requested throughout the manuscript, with changes marked in red text.

Reviewer’s Comment: 1) The introduction presents a coherent and relevant review of previous studies, effectively identifying a gap in the existing literature. The focus on the need for longitudinal or prospective designs for a deeper evaluation of habit theory is crucial. However, although the influence of different forms of consistent cues on the development and enactment of habits is mentioned, it would be useful to specify which forms these cues will take and why they are expected to have a differential impact.

Author’s Reply: We thank the reviewer for this valuable comment and agree that a discussion on this would be useful. While currently there is limited research in this regard, we have expanded upon our discussion of this topic within the introduction on Page 2:

“Despite theoretical assertions of the potential role of cue consistent contexts in the development of habits, empirical evidence supporting such effects is limited. For example, a seminal study in habit development found cue self-selection for a health behavior was associated with the development of stronger habits for that behavior over time [16], yet since this study there has been a dearth of research investigating the effects of different forms of cues on habit development and enactment. Such a contention may be particularly important, as some habit theory has speculated that different forms of cues may have differential impacts on how likely they are to result in the development of habits [19]. That is, such research suggests cues that are more likely to be salient and easily noticed (e.g., going for a walk after dinner) should be more likely to result in the development of habits than cues that are less salient or require conscious monitoring or input (e.g., going for a walk each day at 6pm requires a person to be consciously aware of the time).”

We also expand upon this in the Discussion section, for example on Page 7:

“Results in this regard were partially in line with expectations, as the indirect effect of past behavior on prospectively measured behavior via habit was significantly stronger in participants reporting engaging in physical activity at the same time, as the same activity, and in the same mood. However, in contrast to our hypotheses, engaging in physical activity in the same place, with the same people, and at the same part of one’s routine did not significantly moderate the mediating effect of habit, despite effects in the expected direction. These effects present a unique contribution of this research, indicating that while cues are likely an important element in promoting habit development and enaction, not all forms of cues appear to have equal impact in this regard. While some differences between the role of cues was expected, current findings may be of particular interest as, from habit theory, one may expect more salient cues (e.g., routine) to be stronger mediators of this effect than cues such as time or mood [19,20]. Yet, the current results do not indicate this to be the case, as time but not routine moderated the effects of past behavior via habit on physical activity.”

Reviewer’s Comment: 2) I appreciate the use of PLS-SEM to analyze the structural relationships in the study, which is suitable given the nature of the hypotheses posed. However, it is crucial to include details about the reliability of the scales used to measure each construct. This would not only increase the transparency of the study but also allow readers to evaluate the robustness of the results. I recommend providing reliability coefficients such as Cronbach's alpha, Jöreskog's rho, or McDonald's omega for each scale used. Additionally, it would be beneficial to discuss construct validity through convergent and discriminant validation, providing values for factor loadings, average variance extracted (AVE), and the maximum shared variance (MSV) or the square root of the AVE compared with inter-construct correlations. Including these methodological details would not only enrich the methodology section but also strengthen the credibility of the study's findings.

Author’s Reply: We agree with the reviewer on the value of scale reliability measures. We have now included both Cronbach’s alpha, the AVE, and square root of AVE in Table 2. Notably, the square root of the AVE for each construct was higher than all inter scale correlations. We also note in the results section on Page 4 that all items loaded significantly onto their respective factors (all p values <.001). For completeness, we now also include a link to an associated OSF page, which includes a complete set of all model outputs, including factor loadings and reliability statistics for all iterations of the multi-group model.

Reviewer’s Comment: 3) Moreover, I note a significant omission in the lack of reporting of model fit measures. It is essential to include fit indices such as the Normative Fit Index (NFI), Chi-square (Chi²), and the Standardized Root Mean Square Residual (SRMR), among others. These indices are fundamental to evaluating the adequacy of the proposed measurement and structural models, thus allowing readers and reviewers to judge the quality and reliability of the obtained results.

Author’s Reply: We agree. While SmartPLS does not provide a NFI or Chi2 for bootstrapped models, we have included the SRMR for each model within Table 3. All models showed good fit to data, with SRMR values from .029 to .057.

Reviewer’s Comment: 4) In the methodology section, the use of PLS-SEM to examine the structural relationships between latent variables is detailed. However, to improve clarity and comprehensibility of the study for readers, I strongly recommend that the authors include a visual graph of the proposed structural relationships. A diagram showing the relationships, including path coefficients and significances, would facilitate the interpretation of the models and allow readers to more effectively visualize the dynamics between the variables. Furthermore, this type of graphical representation is a standard practice in the presentation of results in studies applying SEM and would be a valuable addition to complement the textual description of the analysis.

Author’s Reply: Thank you for this suggestion. We have now added a figure denoting the model into the manuscript on Page 5, Figure 1.

Reviewer 2 Report

Comments and Suggestions for Authors

A study conducted among college students in the US aimed to investigate the relationship between habits and physical activity and the effect of consistent stimuli on the development of physical activity habits. The results indicated that habit plays an important role in predicting future behavior, and that its influence is stronger in people who engage in physical activity at consistent times, who exhibit consistent mood and who perform the same activity. In contrast, a fixed place, company and part of a routine had no significant effect on the development of habits. These results support habit theory, suggesting that constant contextual and internal stimuli are crucial for the development of habits and their activation in the future.

The study included 68 students who completed questionnaires about their physical behavior and physical activity habits. The study makes an important contribution to understanding the role of habits in physical activity and confirms the importance of consistent stimuli for habit development. 

They suggest improving the tables to be more readable, they need proper formatting, the caption should look 0.00 instead of .00.

In addition, all elements included in the tables should be described in the footer, e.g., "*" The notation Mean and SD can be written on one lineus tosay X ± SD.

Tables 2 and 3 are arranged in the wrong place in the text.

Replace the notation "p Difference" with "p-value".

Table 2 is not understandable, what do the columns mean ?

If you give values in Table 3 indicate "n (%)". or other notation.

Author Response

Reviewer 2

Reviewer’s Comment: A study conducted among college students in the US aimed to investigate the relationship between habits and physical activity and the effect of consistent stimuli on the development of physical activity habits. The results indicated that habit plays an important role in predicting future behavior, and that its influence is stronger in people who engage in physical activity at consistent times, who exhibit consistent mood and who perform the same activity. In contrast, a fixed place, company and part of a routine had no significant effect on the development of habits. These results support habit theory, suggesting that constant contextual and internal stimuli are crucial for the development of habits and their activation in the future. The study included 68 students who completed questionnaires about their physical behavior and physical activity habits. The study makes an important contribution to understanding the role of habits in physical activity and confirms the importance of consistent stimuli for habit development.

Author’s Reply: We thank the reviewer for their time and effort in reviewing our manuscript, and for the valuable comments. We have addressed each of the concerns raised, with any changes marked in the manuscript with red text.

Reviewer’s Comment: They suggest improving the tables to be more readable, they need proper formatting, the caption should look 0.00 instead of .00.

Author’s Reply: We agree with the reviewer of making edits to ensure the readability of the tables. Each table now has an expanded note detailing its contents, and Table 3 has been reformatted to more clearly allow for readers to understand its contents. As for the use of leading zeros in tables, we originally followed the APA system, in which numbers which are bound to below 1 do not include leading zeros. We now include these in tables, but defer to the editor for their preferred formatting of numerical data, and are happy to add or remove these as required.

Reviewer’s Comment: In addition, all elements included in the tables should be described in the footer, e.g., "*" The notation Mean and SD can be written on one lineus tosay X ± SD.

Author’s Reply: We have included the following in Table 2 to explain the meaning of *, **, and ***: Note. * indicates p < .05, ** indicates p < .010, and *** indicates p < .001.

Reviewer’s Comment: Tables 2 and 3 are arranged in the wrong place in the text.

Author’s Reply: We agree and have moved the tables to be closer to their relevant discussion points in the manuscript.  

Reviewer’s Comment: Replace the notation "p Difference" with "p-value".

Author’s Reply: We have replaced p difference with “p value of difference” in Table 3, and provided a note beneath the table explaining this value: “Note. p value of difference refers to the p value for the bootstrapped difference test for the strength of parameter estimates between those with a consistent or inconsistent cue.”

Reviewer’s Comment: Table 2 is not understandable, what do the columns mean ?

Author’s Reply: We have now added an additional note to Table 2 to aid in the readability of the Table: “Numbers in the column header row refer to each of the numbered constructs within row names”.

Reviewer’s Comment: If you give values in Table 3 indicate "n (%)". or other notation.

Author’s Reply: We agree and have reformatted table 3 to more clearly reflect that the statistics provided reflect parameter estimates and p values for each multigroup model.